# Augmenting Industrial Control Rooms with Multimodal Collaborative Interaction Techniques

Jessica Rubart *, Valentin Grimm and Jonas Potthast

Department of Environmental Engineering and Applied Computer Science, Ostwestfalen-Lippe University of Applied Sciences and Arts, 37671 Höxter, Germany; valentin.grimm@th-owl.de (V.G.); jonas.potthast@th-owl.de (J.P.)
* Correspondence: jessica.rubart@th-owl.de

**Abstract:** The German manufacturing industry has been carrying out new developments towards the next industrial revolution, focusing on smart manufacturing environments. Our work emphasizes human-centered control rooms in the context of production plants. Increased automation does not have to come with less human control. Therefore, we report on multimodal collaborative interaction techniques to augment industrial control rooms. In particular, we include mobile workers who use the control room while being in the production hall using tablets or specifically mixed reality glasses. Collaborative annotation dashboards support discussions and a shared understanding among analysts. Manufacturing-related data can be integrated into business analytics environments so that holistic analyses can be performed. Multimodal interaction techniques can support effective interaction with the control room based on the users' preferences. Immersive experience through mixed reality-based three-dimensional visualizations and interaction possibilities support users in obtaining a clear understanding of the underlying data.

**Keywords:** control room; multimodel interaction; augmented reality; mixed reality



## 1. Introduction

The German manufacturing industry has been carrying out new developments towards *Industry 4.0* [1]. This fourth industrial revolution has introduced the Internet of Things (IoT) and services into the manufacturing environment. Machinery, warehousing systems, and manufacturing are going to be globally connected, which enables smart factories and smart assistance systems. The IoT industrial revolution has contributed to the big data trend in industry. An increased interest for data analytics in manufacturing has emerged. Powerful predictive models are being developed, e.g., for predictive maintenance or predictive quality. This paper presents our augmented digital control room, which extends previous work on a digital control room.

Control rooms support the monitoring and control of complex processes [2]. They are used in numerous industrial settings, such as traffic control [3], emergency services [4], power plants [5], or production plants [6]. Typically, ongoing processes are monitored, and possible incidents or problems are investigated and solved to keep the systems running effectively and efficiently. Moreover, process-related data are analyzed to learn from the different processes and optimize them.

In this paper, we focus on production plants in the context of Industry 4.0. The vision for this industrial revolution focuses on autonomous systems that are able to control and configure themselves to different situations [1]. However, increased automation does not have to come with less human control [7]. Shneiderman proposes a two-dimensional framework of Human-Centered Artificial Intelligence. It differentiates human control from computer automation and opens up a design space for intelligent systems that are reliable, safe, and trustworthy and, in turn, widely accepted. From this perspective, we propose multimodal collaborative interaction techniques to augment an industrial control room.

Our augmented digital control room includes technologies for industrial analytics, e.g., data integration or intelligent algorithms to identify and classify anomalies or predict maintenance tasks. However, in particular it focuses on integrating innovative interaction techniques, such as collaborative and contextual dashboards, speech and gesture control, as well as mixed reality (MR) smart glasses. Multimodal interfaces provide users with a number of different ways of interacting with a system. This might relate to the usage of diverse devices or to natural modes of communication, such as speech, body gestures, or handwriting [8]. The augmented digital control room, presented in this paper, is built upon the digital control room Section 3.2, which in turn is built upon the digital boardroom Section 3.1. Users of the digital boardroom have different situation-dependent preferences with respect to the mode of interaction [9]. Thus, it is important to provide users with different interaction possibilities. Multimodal interaction techniques support the access of the same functionality in different ways depending on the current situation [10]. MR devices, such as the Microsoft® (MS) HoloLens 2, support an immersive experience combining the real world with the virtual one. For example, three-dimensional holographic visualizations can be used to improve the human perception and understanding of the underlying data. We propose contextual dashboards, such as ones presenting the workload of a nearby machine, three-dimensional holographic visualizations, such as scatterplots, and eye-supported details on demand.

In addition, collaboration support is crucial. Industry 4.0 usage scenarios also point out "networked manufacturing" [1], because of which dynamic networked organizations are important. They are based on distributed cooperating teams who need support for the development and execution of joint business processes, which are highly variable [11]. Furthermore, decision making is inherently a collaborative task [12]. Well-founded decisions require the participation of several analysts, such as domain experts, line-of-business managers, customer representatives, or key suppliers. Thus, our augmented digital control room includes collaboration facilities for both—remote as well as face-to-face situations. In particular, we include mobile workers who use the control room while being in the production hall using tablets or MR glasses. With the help of Microsoft Shared Experience, for example, holograms and dashboards can be shared and interacted with collaboratively.

The remainder of this paper is organized as follows. In Section 2 related work is presented and discussed. Section 3 presents our augmented digital control room with special focus on multimodal collaborative interaction techniques. Thereafter, we present our study and results, followed by a discussion. The paper ends with conclusions and future work.

## 2. State of the Art

In the following, related work is presented in the areas of business analytics as well as visualization and interaction techniques for control rooms.

### 2.1. Business Analytics

Business Analytics focuses on data-driven decision making [13]. Data of the past is analyzed to support current and future decisions. Available tools and environments for business analytics, such as IBM® Cognos®, IBM® SPSS Modeler, RapidMiner®, or Microsoft Power BI®, mainly address data integration, analysis, partly including strong machine learning algorithms, and standard visualizations on desktop computers. However, multimodal collaborative interaction techniques are not in the focus.

For MS Power BI, a first app integrating the HoloLens 2 device exists. It supports viewing two-dimensional dashboards, which have been created using the Power BI desktop application or the Power BI online web application. For this, the dashboards need to be available in the MS Azure cloud. Advanced interaction techniques are not yet part of this app.

The SAP® Digital Boardroom [14] focuses on real-time views of business performance. A boardroom traditionally means a room for an organization's board to conduct its meetings. The underlying database technology is SAP S/4HANA®, which concentrates on flexibility

and scalability. From the user interface point of view, three touch screens are proposed to provide an overview, exploration, and context information. The SAP Digital Boardroom could benefit from additional visualization and interaction techniques, such as presented in this paper.

## 2.2. Visualization and Interaction Techniques

Touch and tabletop technologies have been proposed for different application areas, such as computer games [15] or project management [16]. Compared to traditional user interfaces, richer user experience is pointed out. In [5] a study in the context of the "Affordance table" is presented. It focuses on hand gestures for multitouch tables used for monitoring and controlling the processes in a power station environment. The study is taken from a single user perspective, e.g., how many fingers are preferred for a gesture.

In [2] not only touch-based controls are proposed for a control room, but also ones based on tangible objects. They can support the understanding of the controls and give haptic feedback and involve motor skills. Those controls can nicely be combined with interaction techniques presented in this paper.

A multimodal warning display is presented in [6]. It combines auditory and visual information and proposes a special display design for the lunchroom of the operators. A study has been conducted in the context of a paper mill control system.

Generally, presenting data in a form which supports the user becomes increasingly more urgent because there are more and more data, but human cognitive capacity stays the same and slowly becomes the limiting factor (cpr. [17]). This general trend can be countered in various ways.

In this thesis [18] several ways to visualize data are presented to reduce an operator's visual load, in particular with regard to large amounts of data. In addition, concepts of tangible and tactile interfaces as well as adaptive systems are described for a control room.

## 3. The Augmented Digital Control Room

The proposed *augmented digital control room* builds upon our digital boardroom—a multi-display environment, which integrates multitouch and multiuser-based annotation dashboards so that decision makers gain insights into and can discuss an organization's data [19]. Therefore, firstly the digital boardroom is described in a nutshell. Secondly, we give an overview over the digital control room, which integrates manufacturing-related data so that holistic analyses can be performed [20]. In particular, real-time or near real-time data can be visualized and interacted with through annotation dashboards. For this, we have designed a special history functionality. Thirdly, our approach addresses mobile workers who use the control room while being in the production hall using tablets or specific MR glasses. The latter can provide mobile workers with contextual dashboards, three-dimensional visualization and interaction possibilities as well as collaboration facilities while leaving the hands free for manual activities on the machines.

Figure 1 illustrates our augmented digital control room. While previous controllers use one or more screens at their workplace (on the right), mobile workers, in particular those using MR glasses (on the left), can move around the production hall, use contextual information, visualizations, and interaction possibilities, check the situation on-site while having the hands free for manual activities, and collaborate with previous controllers or other peers.

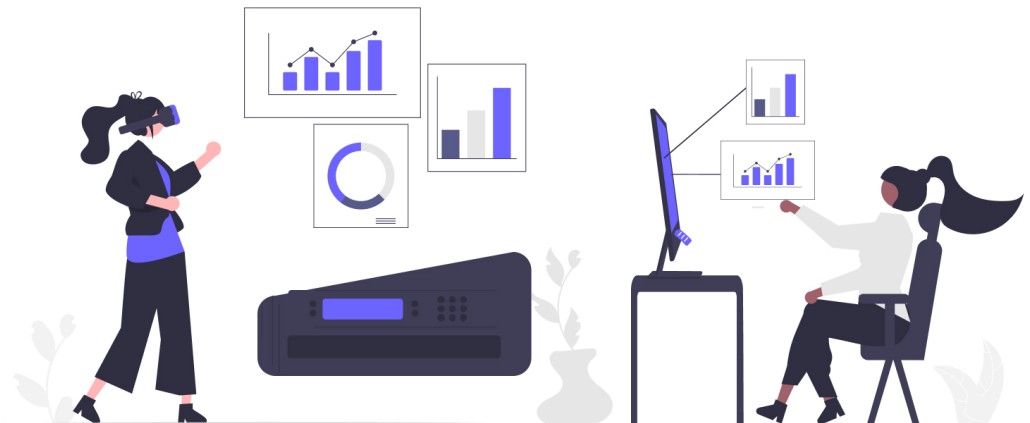

**Figure 1.** The Augmented Digital Control Room: Multimodal Collaborative Interaction Techniques.

### 3.1. The Digital Boardroom

As a joint effort of the research partners Next Vision GmbH and Weidmüller Interface GmbH & Co. KG as well as the Ostwestfalen-Lippe University of Applied Sciences and Arts, we have designed the digital boardroom, which focuses on business analytics integrated in smart meeting rooms [19]. Figure 2 shows one configuration of the digital boardroom. Three multitouch-enabled monitors constitute a shared display space. Of course, this space is not limited to three monitors, but this fits nicely to Shneiderman's Visual Information Seeking Mantra: Overview first, zoom and filter, then details-on-demand [21].

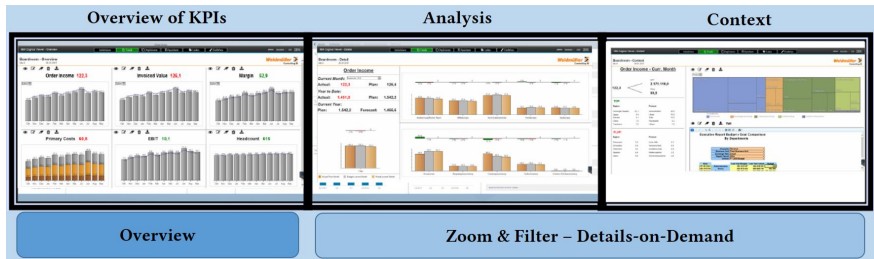

**Figure 2.** The Digital Boardroom: A Configuration.

Therefore, in this configuration the first monitor on the left of the figure gives an overview about a company's relevant key performance indicators (KPIs) presenting their development over time. Each KPI is visualized in a special tile that can be decoupled out of the layout, resized, and annotated. Each tile is implemented as a multitouch object so that gestures can be used for the different interactions. This can be done by one or several users simultaneously as shown in Figure 2. Annotation visibility can be turned on and off as needed. The second screen shows the result of a *drill down* operation on one of the KPIs. In the OLAP (*Online Analytical Processing*) sense this means that a more granular view of the multidimensional data is shown [13], e.g., how *income* as KPI is divided among specific regions. The third screen on the right shows further details on the selected KPI, e.g., value drivers as well as additional contextual information. For example, we integrated a software component for planning purposes and "what-if" analyses based on the multidimensional data.

Multitouch-enabled displays can be arranged in a room, e.g., to constitute an interactive wall. Another possibility is to use such a display as a meeting table (cf. Figure 3 on the right). Meeting participants can interact with the multitouch screens collaboratively. By using the proposed annotation-enhanced dashboards participants are supported in discussing and obtaining a shared understanding of the business data. The first prototype was based on the Business Intelligence (BI) suite IBM Cognos® Analytics 11 as well as HTML 5 and JavaScript for the multitouch and multiuser interaction.

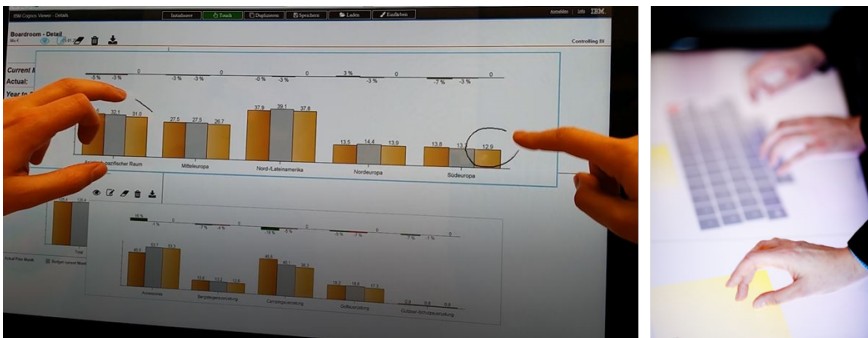

**Figure 3.** Multitouch and Multiuser Interaction.

In collaboration with the company Next Vision GmbH, we have extended the digital boardroom with support for body gestures and speech commands [9]. The prototypical implementation integrates Microsoft Kinect® to identify body gestures and Microsoft Cortana® to recognize speech commands. First interviews with students revealed that for KPIs it has been easy for the interviewees to name potential speech commands whereas for switching screens, navigating through charts or through the multidimensional data model, there was a tendency to propose body gestures, such as moving a hand from left to right. For annotating diagrams, the multitouch features have been the preferred ones. In general, a multimodal environment supporting a personalized way of interaction is helpful.

### 3.2. The Digital Control Room

In another project in collaboration with the company Next Vision GmbH and the HSW University of Applied Sciences, we have introduced the digital control room [20]. The digital control room aims at supporting the monitoring of a shop floor, predictive maintenance of the equipment, and optimizing the production. It is based on the digital boardroom and extends it by integrating the Open Platform Communication Unified Architecture (OPC UA) protocol, which is a widely recognized standard for interoperability and data exchange [22]. The OPC UA integration communicates with an OPC UA server through a publisher/subscriber pattern, cf. Figure 4. Several machines can be connected to one OPC UA server. Through this protocol, we have introduced a real time data source, which we additionally use to store the manufacturing data in a special NoSQL time-series database [20].

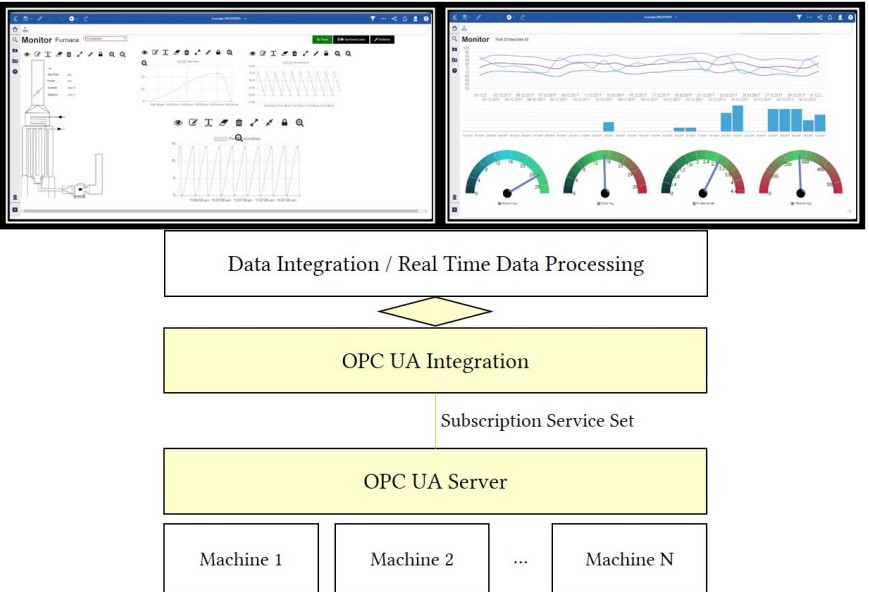

**Figure 4.** The Digital Control Room: Integrating the OPC UA Protocol.

In this way, operational as well as manufacturing data can be integrated in the digital control room and holistic analyses can be performed. In particular, real-time or near real-time data can be visualized and interacted with through annotation dashboards. For this, we have designed special history functionality for the user interface, which allows users to scroll in a time series of data and annotate it with respect to the time dimension. Human-created semantic annotations are captured as an additional data source.

### 3.3. Augmenting the Digital Control Room

In a joint project with the companies flexis AG and Quimron GmbH we have been working on the integration of MR technology in production scenarios. This is the basis for our augmented digital control room. One of the main motivations for the augmented digital control room is its mobility. While a tablet or smartphone can offer an equal mobility to a user, it does not offer the immersive experience where things such as local navigation or information can directly attach to meaningful space in the real world. The scenario we use as a baseline for our concepts is a worker or controller who walks around in a production hall. He or she can see warnings and information of machines and can choose to handle the warnings. Here, a relevant concept for MR software are hand menus that can be used, for instance, to open desired warnings (cf. Section 3.3.2). For solving these warnings and issues of a machine, it is relevant to evaluate past information or upcoming planning schedules. This motivated us to think further about MR data visualization compared to screen-based data visualization (cf. Section 3.3.1) and how to manage these visualizations (cf. Section 3.3.4). In handling warnings, evaluating data and re-planning, collaboration can play a key role. One very relevant kind of collaboration is synchronous remote collaboration where office workers at a desktop PC or other screen-based system interact with the MR user and solve an issue (cf. Section 3.3.3).

#### 3.3.1. Data Visualization

Exploring data is a common task in a digital control room. There are various solutions to display them, e.g., some rely on a tablet you need to hold or some on a big display mounted on a surface. These have in common that they hinder the user in some way or another, e.g., the tablet occupies a hand or the big displays restrict the users' movement. Therefore, although MR hardware solutions such as the Microsoft HoloLens 2 are bulky for the moment, they offer new way to interact with data compared to more traditional devices.

Data Visualization, meaning specifically how data can be visualized and how possible new forms of visualizations may affect data exploration, is a field of active research. There are already studies that deal with this topic. Patrick Millais et al. [23], for example, have investigated whether and to what extent Virtual Reality influences the amount of insights gained by examining graphical information. For this purpose, they created different visualizations in Virtual Reality and also corresponding counterparts in classic 2D visualizations. It was found that the users themselves found data exploration in Virtual Reality more satisfying and successful. The paper emphasizes the motivating character of the representation. Xiyao Wang et al. [24] tried to extend tools designed for data exploration on two-dimensional screens in the context of particle physics with Augmented Reality (AR) features. They stuck to mouse and keyboard input and did not introduce new input devices. Their evaluators found their extension mostly complementing the PC view and recognized the MR view as a useful tool for three-dimensional inspection. However, they also noted that they would still prefer to use two-dimensional tools for some tasks. Concluding their work, they noted a dedicated input for the AR space might be useful. They were surprised that the lack of orthographic projection did not bother their evaluators.

Complementing existing work our research question for this part of this work is:

*RQ 1: Does the immersive experience help to solve the given task?*

### 3.3.2. Interface Design for Menu Navigation

Interface design in a digital control room is a key issue you have to solve in order to create a digital control as described in Section 3.2. In MR, interfaces can be created in various ways. Firstly, there are traditional design elements one can use in desktop applications arranged on a plane in the augmented space. Microsoft provides design templates and most of them follow this design paradigm. Secondly, MR capable devices such as the HoloLens provide in contrast to monitors a three-dimensional space, which enables more immersive and interactive design possibilities. However, interaction is not a goal itself, but should support and motivate the user.

Hendrik Stern et al. [25] have developed two applications, for example, in which they use traditional design concepts for AR. The first one they describe is for mobile phones and uses touch control. The other one is more relevant in the context of this work as it is an application for Microsoft HoloLens. They developed their application using recurring user interviews. Therefore, user feedback was already accounted for in the final evaluation. According to them, it was methodically very important to include user feedback very early in the development cycle. On the design side of things, they found, it is important that the user always knows about which functions and objectives are active. Generally, they concluded, traditional design philosophies can be used as a starting point for making design decisions, but also added there were no design guidelines yet and emphasized further research. Jakub Blokša [26] proposes design guidelines for user interfaces in AR. He recommends that volumetric elements should be used and underlines that spacing is important for visual clarity.

In this work, we use examples to compare traditional design elements and a more immersive counterpart for navigating menu elements. Our research question on this topic is therefore:

*RQ 2: Can three-dimensional elements be used to make the operation of MR applications more intuitive, or is the application of "classical" menus superior?*

### 3.3.3. Immersion and Collaborative Experience

In the context of the augmented digital control room, one interesting subject is synchronous remote collaboration, where users can discuss relevant issues without being physically at the same place. One such situation could arise in the industrial context, when a worker (wearing MR glasses) realizes some issues with a machine and wants to update machine tasks in accordance with the planning office. With respect to the augmented digital control room, this has some special properties compared to the classical digital control room as described in Section 3.2. The immersive experience changes the way of interaction as well as the potential of collaboration compared to the desktop PC. One specifically interesting property is the enhanced ability of creating a feeling of co-presence through the visualization of aspects such as gaze focus and hand visualization. Gupta et al. [27] investigated a setup with an MR user and remote desktop expert and evaluated how different cues in the collaboration process impacted different aspects such as connectedness, fun, and co-presence. Both collaborators communicated through voice chat and the PC user could see a live video from the MR view. The MR user could see the pointer of the mouse position and the PC user could see the gaze focus of the MR user. As a result, these cues increased quality of work, communication and co-presence. Moreover, the gaze focus made the PC user feel the focus of the MR user and increased the fun of collaboration. Recently, Kim et al. [28] published a work, where the MR user collaborated with a VR user. They explored how cues improve the speed, the feeling of co-presence and the mental load. Key findings were that the visualization of a hand-pointer did not significantly affect the task result, while drawing into the field of view improved the speed of work. Both, pointer and drawing, increased the mental load for the users. Our research shall tighten the aforementioned research and includes a similar setting to Gupta et al., but adds visual cues of the MR user's hands and pointers. Moreover, we use the Hololens 2 as a more modern and powerful device for MR applications compared to a hand-crafted device by Gupta et

al. and the Meta2 device by Kim et al., which, for example, does not support eye tracking and does not have a build-in microphone compared to the Hololens 2 [29].

In this work, we focus on the following research question with respect to collaboration in the augmented digital control room.

> *RQ 3: How do visual cues of hands and eyes improve the experience of collaboration between PC user and Hololens 2 user?*

### 3.3.4. Object Management in the Dashboard

Managing windows and objects is important when a dashboard application is used, such as in the digital control room. When interacting and visualizing data in dashboards, it is desirable to replace, remove, and set aside data objects. For example, in one situation a user might want to focus on a single dataset and afterwards he or she wants to find relationships between two different datasets, changing the dashboard frequently.

In the context of MR, challenges such as a moving field of view and an effectively unlimited screen must be overcome. To make this process intuitive, Microsoft PowerBI [30] offers a toolbelt in the context of MR. With this toolbelt, users can move elements into their view, picking them from a list of views visualized around their hips, as well as moving them back into the belt if they are no longer needed. Testing this, we found that it feels unnatural to move the head down to the belt and one can easily lose orientation of objects placed on the belt. Moreover, the toolbelt can not be moved manually, which can lead to usability issues. To mitigate these issues, we developed our own version of such an object management tool that has a similar shape as a mapstand. The idea is that it is a single movable object consisting of a number of vertically stacked cubes. Every cube is horizontally rotatable, potentially holding one object on each side that can be removed and plugged back in. By giving users the ability to add or remove cubes from the mapstand, they can increase or decrease its size as needed. If desired, a cube can even hold an arbitrary number of objects with an infinite scroll comparable to the rotation of the rotatable dice menu.

For the rotatable dashboard we focus on the following research question.

> *RQ 4: Can a three-dimensional holographic object help to organize dashboards compared to a layout based organization?*

## 4. The Study

In order to receive usage feedback on the design space for our augmented digital control room, we conducted a study based on four different prototypical applications for the HoloLens 2 device and one for the desktop PC. The study focuses on the research questions, motivated and phrased in Section 3.

In the following, we describe the four different experiments, which include the setting, a description of the application(s), and the tasks for the subjects, respectively. The applications are illustrated in the context of the *SmartFactoryOWL*, the industry 4.0 lab in Ostwestfalen-Lippe [31]. Afterwards, we present the results of the study and discuss them in the context of our augmented digital control room.

### 4.1. Experiments

For the purpose of answering the aforementioned research questions, we developed four prototypical applications, which were tested and assessed by individuals older than 18 years, independent of gender, technical experience and other factors. Beyond age, the only relevant property of the individuals was that they are physically able to solve the tasks. Most participants were students of ages between 20 and 30 years. Every participant, if possible in the given time slot, evaluated all four prototypes sequentially. Because of dense time frames of some participants the number of participants varies between 13 and 16. After evaluating each prototype, participants answered single-choice questions and one question that offered room for free comments on the application regarding positive, as well as negative aspects. In addition to the questionnaire feedback, we were open to verbal feedback and general sentiment throughout the evaluations.

### 4.1.1. 3D Immersive Scatterplot

To address RQ 1, we have selected a 3D interactive scatterplot for our experiment as well as a simple data set representing cars and their performance indicators. The cube-shaped scatterplot consists of three axes and several points located in the 3D coordinate system. The points represent the cars. The dimensions of the coordinate system are bounded by the axes. Each horizontal dimension is assigned a performance characteristic in the first scenario, while the vertical dimension shows the price dimension. The origin of the axes is not zero. Instead, each axis has as its minimum value set to the lowest value of the respective characteristic from the set of cars. The maximum value is consequentially the largest value. Outside of the scatterplot itself are two smaller cubes located to manipulate the transformation of the scatterplot. In Figure 5b, the blue plane shows the performance indicators of the selected car/point.

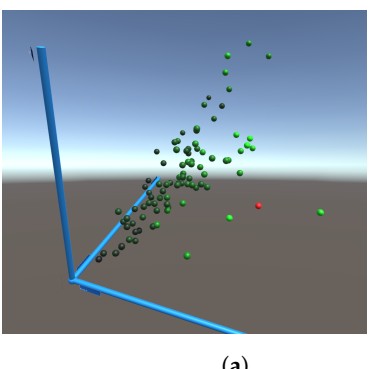
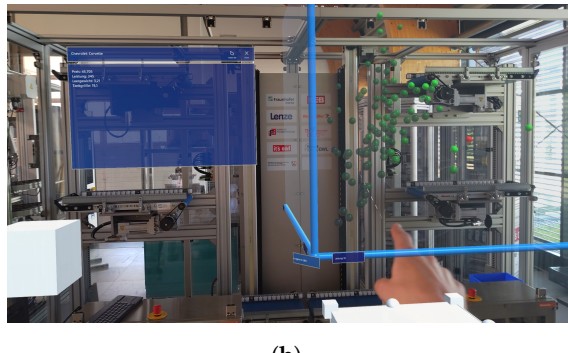

|(**a**)|(**b**)|

**Figure 5.** View of the scatterplot applications used in the experiment. (**a**) Desktop application; (**b**) HoloLens application.

The scatterplot is partly a self-developed solution. Some premade scripts are used, provided by the Mixed Reality Toolkit from Microsoft. For example, for constraining the transformation information of the two cubes and the scatterplot, premade scripts are used. The scatterplot itself is built with functions provided by the Unity Game Engine.

Each car is defined by its manufacturer and model name and has a price, horsepower, curb weight, and tank size associated with it. The evaluation is split into two scenarios. In both of them, the user is supposed to find the best car in terms of performance per price using the scatterplot. In the first scenario (3D scenario) the user has to take into account the price dimension and two performance characteristics. Performance indicators are horsepower, curb weight or tank size. In the second scenario (4D scenario), the task is the same as in the 3D scenario, but performance indicators are also assigned to the vertical axis and the price dimension is visualized by coloring: the brightest car being the most expensive and the darkest the cheapest. By comparing the cars selected by the users with a list of cars sorted by their equally weighted price/performance ratio the evaluation is in parts quantifiable. Test persons were asked to use both—the 3D interactive scatterplot with the HoloLens 2 device (cf. Figure 5b) and an equivalent application for a desktop pc/laptop with adapted controls (cf. Figure 5a). Both applications are based on the Unity framework. Test persons alternately used the Desktop or the HoloLens application first to reduce skewing of the evaluation results through learning effects. This enables us to compare the performance of the test persons with respect to the different devices.

Figure 5b shows the scatterplot in front of the smart warehouse demonstrator of the SmartFactoryOWL [32]. In this industrial context, the scatterplot could be used to position the different components, such as flat conveyers, in the space of key performance indicators, such as power consumption, speed, or voltage.

In order to answer RQ 1, we asked the following questions to the test persons:

- *Which of the two applications made the decision process easier for you?*
- *Which scenario made it easier for you to orientate yourself?*

- *In which of the two scenarios was it easier to include the price information indexed by the colour in your decision making?*

### 4.1.2. Dice Menu for Navigation

To compare traditional design elements with a more immersive counterpart for navigating menu elements (RQ 2), we have prepared a setting with three different kinds of hand menus (cf. Figure 6). A menu floating in the space in front of the user allows them to switch the active hand menu. The different variations share a common position on the user's lower arm. Each particular hand menu is described in detail below.

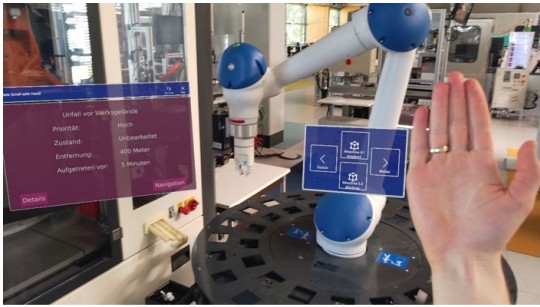
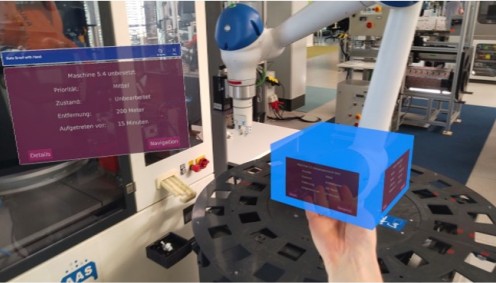
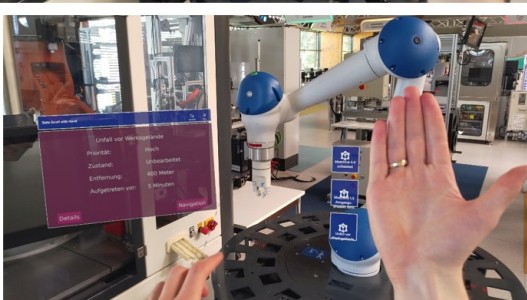

**Figure 6.** View of the three variations for our hand menu evaluation. Upper left: click menu; lower left: scroll menu; right: cube menu.

The rotating-cube menu is a special 3D hand menu we have designed and implemented prototypically. Its main design element is a three-dimensional cube. The cube floats on top of the user's hand and can be rotated around its y-axis using different hand positions. In our industrial setting, the controller can navigate through warning messages, for example. While rotating, the warning messages are continuously exchanged on the side of the cube facing away from the controller. This allows the cube to display more than four elements in its function as a navigation element. To show the cube, the application needs to recognize that the user holds his or her hand flat, and to rotate the cube, the user has to lift her or his thumb.

The scrollable list menu mostly uses premade elements from Microsoft's Mixed Reality Toolkit, e.g., a *GridObjectCollection* arranges the buttons automatically in the needed structure and a *ScrollingObjectionCollection* handles the user input. When the user points a finger at the list, she or he can move the list up and down by moving the finger.

The click menu displays content blockwise using a tile-based layout. The user can navigate to the next or previous content block by using a forward or a backward button, respectively. It is technically the simplest solution.

In this scenario, the test person is asked to navigate through a set of warning messages and select two given ones in succession. In Figure 6 the hand menus are shown in front of a robotic arm in the SmartFactoryOWL. The warning messages can refer to this industrial demonstrator. The test person has to repeat the selection of the given warning messages three times using the previously described menus, each containing the same content.

In order to answer RQ 2, we asked the following question to the test persons:

- *Which menu was the most user-friendly to you?*

### 4.1.3. Synchronous Collaborative Planning

To test RQ 3, we created a collaborative prototype application featuring a reasonable use case in the context of our project. The idea is that one worker in the production hall updates the task planning together with one planner from the office. The application view of the desktop user is shown in Figure 7.

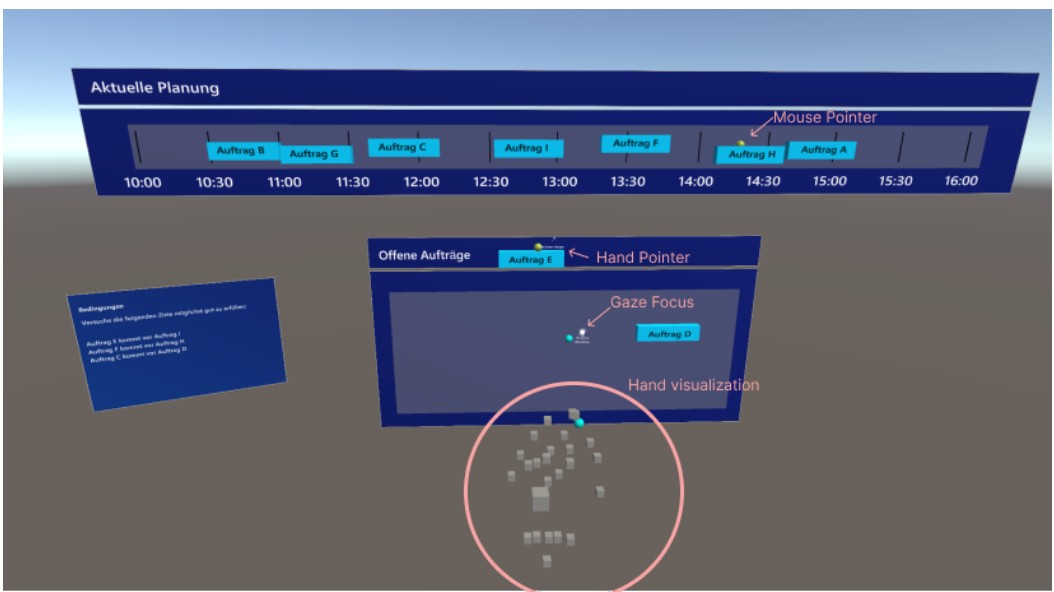

**Figure 7.** View of the collaborative application from the perspective of the desktop user.

The interaction surface of the user interface is visually the same for the desktop and HoloLens user, which consists of two slates: one is the slate "current planning" which shows a timeline, where tasks can be added into time slots. The other slate contains the tasks which could not be allocated to a time slot yet. Tasks can be moved around via drag-and-drop by mouse (desktop user) or by hand-ray (HoloLens user). Releasing a task outside of the slates pushes it back to the initial position before dragging.

To assess RQ 3, we set up two experimental settings: in one (setup 1) we display cues of co-presence for both users, and in the other no cues are visible (setup 2). In setup 1, the desktop user can see the hands of the HoloLens user, as well as the hand-ray, i.e., the extension of the finger pointer that can be used for far-interaction, as well as his gaze focus. The positions of the visual cues are retrieved with the help of the Mixed Reality Toolkit (MRTK) from Microsoft. The HoloLens user can see the mouse pointer of the desktop user.

The synchronization between HoloLens and desktop app is done with the help of Photon Unity Networking 2, using their free server resources and the Photon.Pun library. Here, a lobby is created on startup from the HoloLens app and all elements are synchronized with their position, rotation and scale for the desktop app. Photon uses an ownership system, where elements created by one user, are usually owned by that user and can only be moved by him or her. To give both users the ability of moving tasks around, remote procedure calls are used.

A pair of users (collaborators) is needed to conduct this experiment. One user uses the desktop application with the mouse and the other user uses the HoloLens application. The collaborators can communicate with each other by voice chat. As described in the previous section, there are two alternative experimental setups evaluated: in one setting, visual cues of the users' movement are shown to each other, whereas the other setting omits those cues. The cues visible to the desktop user are the hands, the focus point of the view, and the pointer from the hand of the HoloLens user. The HoloLens user can see the position of the mouse on the task planning surfaces. To antagonize the learning effect happening in the first run, potentially skewing the results of the second run, we swap the starting setting between each evaluation. Both users receive independent conditions that

they have to satisfy in the task planning task with the form <Task A> comes after <Task B>. If it is not possible to satisfy the conditions for both users, the collaborators have to find an optimal compromise. This means that the deviation of satisfied conditions shall not be more than one. In order to answer RQ 3, we asked the following questions to the Desktop user:

- *How much did the visualization of the hand of your collaboration partner help you to solve the task?*
- *How much did the visualization of the view of your collaboration partner help you to solve the task?*

The HoloLens user was asked:

- *How suitable are Holograms and control via hand gestures for the task?*
- *How much did the visualization of the mouse of your collaboration partner help you to solve the task?*

### 4.1.4. Rotatable Dashboard for Object Management

In order to address RQ 4, whether a 3D holographic object can help to organize dashboards, in particular improve a layout based organization, we have designed a holographic object, which we call a *mapstand* or *Rotatable Dashboard*. It is meant to be a proposal for a different visualization technique providing overview over different interfaces designed on two-dimensional planes. Technically, it uses primarily assets provided by Microsoft in the Mixed Reality Toolkit. It consists of multiple cubes, which are aligned alongside a cylindrical object. The cubes can be rotated around the vertical axis of the cylinder. The slates, which are two-dimensional planes provided as assets in the Mixed Reality Toolkit, represent potential interactive displays or dashboards that can be used for a variety of tasks.

In this experimental setup, the user tests and compares the two applications (cf. Figure 8). On the one hand the toolbelt of Microsoft PowerBI and on the other hand our mapstand. This is again done so that the first application is swapped for every evaluation. The user is instructed to take elements from the toolbelt/mapstand and arrange and rearrange them in their field of view as they like. Figure 8 shows the two alternatives in front of the smart warehouse demonstrator of the SmartFactoryOWL [32]. Such dashboards can be used to show various analyses of this demonstrator.

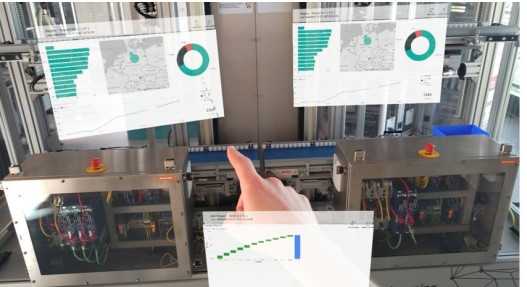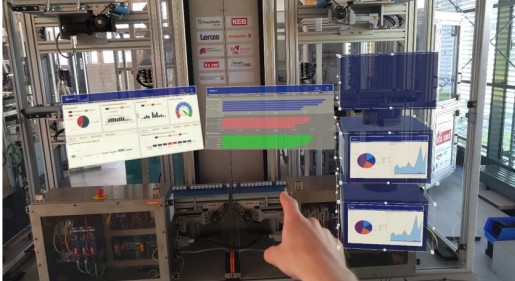

**Figure 8.** **Left**: View of the Microsoft PowerBI Toolbelt application; **Right**: View of the rotatable dashboard/mapstand.

In order to answer RQ 4, we asked the following question to the user:

- *Which of the two applications do you think are more user friendly?*

### 4.2. Results

Figure 9a shows the results of the survey. Most of the participants thought that the HoloLens application made the decision-making easier for them than the other application. Furthermore, most of them found that it was easier to orientate in the three-dimensional data using the HoloLens. It was possible for the participants to give optional form-less feedback. In that, some suggested using a multi-colored gradient because it was difficult for

them to differentiate between the various shades of green. This is consistent with feedback we received in the questions as shown in Figure 9a. Furthermore, most of them found it helpful to be able to move around to explore the data. There were some suggestions to improve the HoloLens application. Some found that positioning using the cubes was suboptimal and suggested edge-bound controls. Additionally, some noted that the axis-description was not always easily readable.

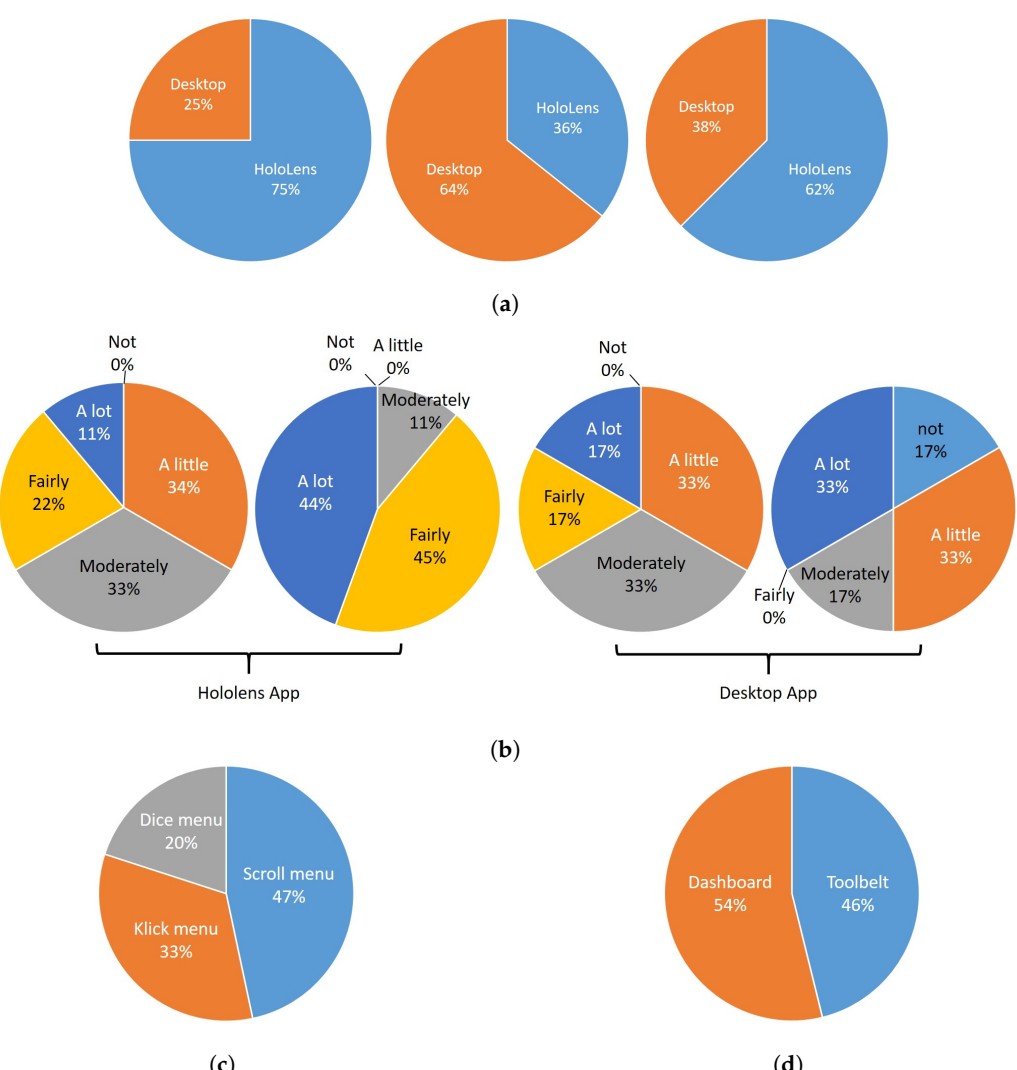

**Figure 9.** Results of the experimental evaluation for the four different experimental setups ((**a**)–(**d**)). (**a**) Shows the results for the evaluation of the scatterplot application. Left: Easier choice in the 3D scenario; middle: Easier choice in the 4D scenario; right: Overall better orientation; (**b**) Shows the results for the evaluation of the collaboration application. From left to right: Usefulness of hand gesture interaction for the HoloLens user; helpfulness of mouse visualization for the HoloLens user; helpfulness of hand visualization for the desktop user; helpfulness of gaze visualization for the desktop user; (**c**) Shows the results for the evaluation of the handmenu application. The pie chart displays which menu the participants rated as most user friendly; (**d**) Shows the results for the evaluation of the dashboard applications. The pie chart displays which application was rated more user friendly by the participants.

Figure 10 shows the distribution of the answers to the given question alongside the price-performance ratio axis. For the calculation of the boxplots we divided performance values of each car by its respective price. The performance value for each car is calculated by min-max normalization (interval [0, 1]) of the performance indicators and afterwards, calculating the increased efficiency of the selected car compared to the efficiency of the

cheapest car. This is done by using the Euclidean distance. Therefore, the higher the values in the boxplots the better the car selection.

**Figure 10.** An overview of the cars selected by the evaluators in the different scenarios and their price-performance ratios from the scatterplot scenarios.

The boxplots for the 3D scenario show similar results for HoloLens and desktop application. Picks with the HoloLens were insignificantly better. The results for the 4D scenario show that the picks on the HoloLens were worse. This is most likely related to the decreased color saturation on the HoloLens as described earlier. Therefore, we can not derive a significantly better result from the data for the immersive application compared to the desktop application.

Figure 9c shows which menu was ranked as most user friendly. The participants had the possibility to give formless feedback too. There were some technical issues due to the design of the rotating-cube menu. Some had problems opening and rotating it. Still, they thought the idea to use three-dimensional objects as menu elements was promising. The feedback on the click- and rotating-list-menu were mixed. Some preferred the scrolling, some the clicking.

The results of our evaluation of the collaborative planning app are shown in Figure 9b. Because of the limited amount of time for every evaluation, it was not possible to let the participants take both perspectives from the HoloLens and the desktop app. This led to nine responses for the HoloLens app and six for the desktop app.

The usefulness of the hand gesture interaction from the HoloLens has been rated with a high amount of "a little" and "moderate" usefulness. Participants reported that picking and moving tasks was much slower than on the desktop with a mouse. Additionally, the far interaction did not work well for many participants and needed a good technique to be recognized by the HoloLens properly.

The visualization of the mouse on the other hand, was mostly rated as "quite helpful" or even "a lot" helpful, with one person stating it was only moderately helpful.

The desktop users reported that the hand visualization of the HoloLens user did only help a little to moderately. One participant said it was very helpful. One of the main reasons for this bad rating was the fact that the hand was mostly not visible due to the far interaction used and only the pointer of the hand did show up on the desktop screen. Additionally, due to the problems with moving tasks for the HoloLens user, the effect of the hand pointer being visible was further diminished.

The usefulness of the gaze visualization was rated comparably with one person saying it was not helpful at all, but on the other hand one person more that said that it was very helpful.

The results for the dashboard evaluation are shown in Figure 9d. The question regarding user-friendliness shows controversial results with a slight edge for the dashboard application. Most responses of participants included that they had a better clarity with the rotatable dashboard while some had more clarity with the toolbelt (especially for a



smaller amount of objects). Additionally, written responses included that the rotatable dashboard was easier to use. Opposingly, some participants reported that moving windows was harder with the dashboard-app which led to a worse assessment with respect to its user friendliness.

### 4.3. Discussion

The scatterplot application for the evaluation showed only a basic visualization, but the user feedback indicates that data exploration is a potential purpose for the HoloLens. The feedback we received regarding the cubes is reasonable as the scatterplot can become too large for the field of view of the HoloLens device. An alternative way to provide transformational controls would be to implement them in a special hand menu. Adding edge-bound controls could overlap with controls for additional functions we had in mind. We did not add them to the evaluation to keep the experiment simple. The readability of text on the HoloLens is due to technical limitations. The resolution of the display is only 1920 × 1080 and, when moving away from virtual object,s they become smaller and are reduced in pixel count due to their size. A possible workaround until technical limitations are lifted by more advanced platforms could be to scale font size to some degree with distance to the user.

As already mentioned, we had interactive functions in mind when developing the application originally. For example, we thought about giving the user the possibility to reduce the amount of data viewed by drawing a three-dimensional cube. We discarded the idea in favor of a simpler experiment, but such functions could further immerse the experience for the user. Our application shows only the minimal and maximal values. Further research could try to create a grid for MR-systems. Regarding RQ 1, we recognize the potential of MR-systems for data exploration. Further research could try to introduce a more immersive experience by adding functions to manipulate the visualization or evaluate other diagram types.

With respect to the evaluation of different menu types, the technical problems need to be considered when interpreting the results. As mentioned, while we had problems with the rotating cube menu, most participants still thought three-dimensional objects could be interesting in user-interface design. The mixed results of the evaluation showed all types can be viable. To improve user satisfaction as much as possible, integrating user feedback into the development as early is recommendable. Regarding RQ 2, we can say that none element is superior to the other. Further research could try to integrate different elements with each other and create templates for standard scenarios. In general, a multimodal environment supporting a personalized way of interaction is helpful, e.g., by using dedicated interaction or structure models in the software architecture [9,33].

The results for the collaborative application suggest a limited use of the visual cues of the MR-user for a collaborative task. Moreover, solving this task was reported by many participants to be much easier without the interaction of the HoloLens user. The collaboration with the HoloLens user seemed to be more effective by taking an observing stance, only using voice chat to instruct the desktop user. Showing the movement of the desktop user's mouse seems to be a helpful feature.

Especially for a task that is accomplished on the 2D plane, the desktop mouse is a very effective tool.

Based on this assessment, RQ 3 can be answered as follows: Visual cues of eyes and hands did not significantly improve the experience of collaboration for the planning task. Aside from general usability issues connected to the abilities of the HoloLens 2 itself, there were two main reasons for this: (1) The planning task can be handled much easier from a screen-based device, specifically with a mouse. This makes intervention from the HoloLens less relevant besides voice suggestions. (2) To make good use of the hand visualization, near interaction with the elements is needed. This comes with the trade-off of having to make bigger hand movements and/or decreasing the size of the interaction elements.

Interesting scenarios for further research on the collaboration of MR- and desktop-user could be in a context of modeling in the 3D space, where rotation in the 3D space and

moving around is more relevant. Another interesting scenario for such a collaborative setup is data analysis, where the main use is not dependent on moving parts, but showing/hiding information and leveraging the immersive space.

With respect to the object management tool evaluation, we found that more than half of the participants reported that the mapstand was more user friendly. The results indicate that the rotatable dashboard can be a viable alternative for many. This confirms RQ 4.

Based on the feedback, improved assessment of the conceptual idea of the rotatable dashboard is mainly dependent on upgrading the rotatable dashboard with respect to the ease of moving objects, making elements pop back into the dashboard properly, add the ability to create and delete layers of the map-stand and make it movable, as well as tagging along with the user if needed.

## 5. Conclusions

We propose to augment an industrial control room with multimodal collaborative interaction techniques. Based on previous work on the *digital boardroom* and *digital control room* we have presented our *augmented digital control room*. While previous work focuses on annotation dashboards for supporting collaboration and includes first results on integrating body gesture and speech-based interaction, our augmented digital control room includes mobile workers who use the control room while being in the production hall. This is mainly based on using MR glasses, i.e., the Microsoft Hololens 2 in our studies.

The main findings of our evaluations are summarized in Table 1.

**Table 1.** Overview of the main findings of our studies.

| | |
|---|---|
| RQ 1 | - Measureable perfomance of the participants was on average similiar on both platforms, but from their perspective MR was more comfortable<br>- Color shades on the HoloLens are difficult to distinguish, especially in bright light conditions |
| RQ 2 | - 3D-objects can improve the interface design, but new gestures need to be developed<br>- Perception of what is a good interface is individual, therefore multimodal interaction techniques are crucial |
| RQ 3 | - Having the mouse cursor visualized on the HoloLens improved the satisfaction level for the collaboration<br>- The usefulness of hand and gaze visualization was limited due to the kind of application used |
| RQ 4 | - A significant amount of participants believe that the 3D object serves as a more user friendly alternative<br>- More reliable feedback is dependant on a more developed prototype |

While the Microsoft Hololens 2 is currently one of the best and most relevant MR devices for modern MR applications, it still suffers usability issues such as a small viewport and imperfect gesture recognition. In this study we mostly worked with people that are not trained on such devices. This combination led to the effect that sometimes the concepts that we tried to test with our prototypes were overshadowed by, e.g., gesture recognition issues. Interestingly, it was not the same issue for every participant. Some had issues with the bending of the hand for the dice menu, some had temporary issues with far interaction, while others tended to lose orientation and accidentally opened other apps and windows on the device. Additionally, creating optimal lighting in a room was not always possible and participants suffered fatigue.

The number of study participants was reduced by people that told us about their unwillingness to use the MR glasses due to fear of motion sickness. This assumption shows a general misleading view that virtual reality, which is strongly connected to motion sickness, and MR were the same. Studies show that motion sickness is not a relevant issue for MR applications where the real world is still visible (cf. [34]).

We already have promising results using the Microsoft HoloLens 2 device. However, further development of MR devices is a relevant factor for future research. In our future research, we will focus on extending and improving the MR-based collaboration features, e.g., for collaborative data exploration. More specifically, it is interesting to investigate into collaborative scenarios in this context, where MR users can interact with objects by themselves compared to a situation where only a desktop user can do interactions and the MR user adds to the task by voice and gestures.

We further developed the scatterplot application for a more interactive experience, where users can manipulate the data through zooming or changing dimensions [35]. We plan to evaluate applications such as this in the future because we think that the advantages of the immersive experience will become clearer.

In addition, we would like to investigate further the usage of 3D holographic objects for supporting the digital workspace. On the one hand, we can think of a work desk setup where the mapstand is usually not required to move along with the user because their position is fixed. On the other hand and more generally, it is interesting to further evaluate the mapstand application in a more developed state that eliminates most of the observed usability shortcomings.

**Author Contributions:** Conceptualization, J.R., V.G. and J.P.; Funding acquisition, J.R.; Software, V.G. and J.P.; Writing—original draft, J.R., V.G. and J.P. All authors have read and agreed to the published version of the manuscript.

**Funding:** The German Federal Ministry for Economic Affairs and Energy supported the research studies under grant ZF4132603SS9.

**Institutional Review Board Statement:** All procedures performed in studies involving human participants were in accordance with the ethical standards of the institutional and/or national research committee and with the Helsinki declaration and comparable ethical standards.

**Informed Consent Statement:** Informed consent was obtained from all subjects involved in the study.

**Data Availability Statement:** Data is contained within the article.

**Acknowledgments:** The authors thank their project partners for fruitful discussions as well as Jonas Behler for his contribution to the development of the mapstand.

**Conflicts of Interest:** The authors declare no conflict of interest.

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
