# Peer review of "Augmenting Industrial Control Rooms with Multimodal Collaborative Interaction Techniques"

_futureinternet, doi:10.3390/fi14080224_

Round 1
Reviewer 1 Report
The research article is focused to the augmenting industrial control rooms with multimodal collaborative interaction techniques.
The authors of the research article processed the analytical part of the research activity in a very appropriate way with a subsequent link to the augmented digital control room with special focus on multimodal collaborative interaction techniques with executed experiments and results, followed by a discussion and conclusion.
From my point of view, I recommend the authors to unify the one comprehensive point and not individual experiments divided to the results 4.1.2, 4.2.2., 4.3.2., 4.4.2. and discussion in points 4.1.3., 4.2.3., 4.3.3. and 4.4.3 as each experimental part of the research tasks has its own experiments, results and discussion, but in the end it results in too many divisions and an unnecessary amount of partial sections.
I recommend extended the conclusion and more specify the future research for development of MR devices focus on extending and improving the mixed reality-based collaboration features and collaborative data exploration.
This research area is very interesting and currently highly developing. Modern technologies such as Virtual reality, augmented reality, mixed reality, digital twins are, in my view, the future, and therefore I recommend to accept and publish after the minor revision as it is a little researched area that can bring a lot of research and application potential in the future.
I wish the authors a lot of strength in further research on this issue.
Author Response
Dear reviewer,
Thank you very much for your constructive and motivating review!
Please find our responses to your points below:
“From my point of view, I recommend the authors to unify the one comprehensive point and not individual experiments divided to the results 4.1.2, 4.2.2., 4.3.2., 4.4.2. and discussion in points 4.1.3., 4.2.3., 4.3.3. and 4.4.3 as each experimental part of the research tasks has its own experiments, results and discussion, but in the end it results in too many divisions and an unnecessary amount of partial sections.”
We have restructured our article in the following way:
The subsections 4.1.2, 4.2.2., 4.3.2., 4.4.2., you have mentioned regarding the results of the experiments, are now consolidated in subsection 4.2. Section 4 is about our study and comprises a description about the study, its experiments, the results as well as the discussion. Therefore, the other subsections you mentioned (4.1.3., 4.2.3., 4.3.3. and 4.4.3) are now consolidated in section 4.3.
"I recommend extended the conclusion and more specify the future research for development of MR devices focus on extending and improving the mixed reality-based collaboration features and collaborative data exploration.
This research area is very interesting and currently highly developing. Modern technologies such as Virtual reality, augmented reality, mixed reality, digital twins are, in my view, the future, and therefore I recommend to accept and publish after the minor revision as it is a little researched area that can bring a lot of research and application potential in the future."
We have adapted the descriptions of the experiments with more industrial figures to make this interesting research area more expressive. Furthermore, we extended the conclusions and future work.
All the best,
Jessica Rubart
Reviewer 2 Report
The paper describes how the integration of Mixed and Augmented reality in industrial control rooms can drive toward the envisaged fourth industrial revolution. The German effort in the introduction of new technologies in the industry is cited and described.
The paper cites a lot of sources and the most widespread hardware for Augmented and Mixed Reality are cited and evaluated. Also, the bibliographical review seems to be more than adequate. Strong importance has been given to the interfaces' usability and feedback from users through a set of interviews and tests involving mainly persons from 20 to 30 years old. The paper could provide the reader with insights into the introduction of new technologies in the modern industry.
On the other side, as a reader, I would expect more oriented case studies. The title introduces the application of Interactive techniques in an industrial environment. However, I don’t see in the case studies and in the images industrial backgrounds or practical cases related to the mechanical/manufacturing industries, that is what I would expect from the title. For instance, figure 12 presents an interesting application of video flow augmentation with digital objects, but the background seems to be a bedroom; on the contrary, I would expect a scenario with a factory with CNC or machine tools. Therefore, the paper is well written and interesting, but I suggest you focus as much as possible on the industrial environment, both in pictures both in the case studies. For instance, you could add a picture showing how your interfaces could support the manufacturing of mechanical parts, monitor CNC machines, or how to carry on maintenance on a car or an industrial plant.
Author Response
Dear reviewer,
Thank you very much for your constructive and motivating review!
Please find our responses to your points below:
“On the other side, as a reader, I would expect more oriented case studies. The title introduces the application of Interactive techniques in an industrial environment. However, I don’t see in the case studies and in the images industrial backgrounds or practical cases related to the mechanical/manufacturing industries, that is what I would expect from the title. For instance, figure 12 presents an interesting application of video flow augmentation with digital objects, but the background seems to be a bedroom; on the contrary, I would expect a scenario with a factory with CNC or machine tools. Therefore, the paper is well written and interesting, but I suggest you focus as much as possible on the industrial environment, both in pictures both in the case studies. For instance, you could add a picture showing how your interfaces could support the manufacturing of mechanical parts, monitor CNC machines, or how to carry on maintenance on a car or an industrial plant.”
Due to the COVID-19 pandemic, some of our work took place in home office situations. However, as our university is partner of the “SmartFactoryOWL”, which is a real lab for industry 4.0 in our region (“Ostwestfalen-Lippe”, short “OWL”), we have adapted the case studies and figures with industrial descriptions to make the industrial environment more explicit.
In addition, we have restructured the article to consolidate the results and discussions.
All the best,
Jessica Rubart